# Gender difference in domain-specific quality of life measured by modified WHOQoL-BREF questionnaire and their associated factors among older adults in a rural district in Bangladesh

**Fakir M. Amirul Islam**[1,2]*

**1** School of Health Sciences, Swinburne University of Technology, Hawthorn, VIC, Australia, **2** Organisation for Rural Community Development (ORCD), Dariapur, Narail, Bangladesh

* fislam@swin.edu.au

## Abstract

**Data Availability Statement:** All relevant data is available in the paper and its Supporting Information files.

### Background

The global population of older adults has steadily increased in recent decades. Little is known about the gender difference in the quality of life (QoL) of older adults in the general population. This study aimed to identify factors associated with QoL among older adults by gender.

### Method

Data on QoL using the World Health Organization Quality of Life (WHOQoL-BREF) questionnaire and socio-demographic characteristics, including living status and sources of income, were collected from 1147 older adults. The WHOQoL-BREF has four domains: physical, psychological, social and environmental. Rasch analysis was used to compute a combined score from Likert-type data to a continuous scale ranging from 0% satisfaction to 100% satisfaction in terms of QoL for each domain. We used a generalized linear model to compare the mean rate of QoL for different factors, and logistic regression analysis was used to quantify the associations of factors with below-average QoL measured as 50% or less in QoL.

### Results

The domain-specific QoL mean (standard error), minimum-maximum values were physical 48.9 (0.41), 7–86, psychological 38.9 (0.51), 4–71, social 50.5 (0.49), 8–92, and environmental domains 47.8 (0.37), 6–91 in the total sample with significant gender difference in all but social domain. The proportion of below average QoL for females vs. males was physical 47.6% vs. 42.6%, psychological 74.4% vs.66.7%; social 34.8% vs. 30.1% and environmental domains 56.1% vs. 49.0%. In females, participants living alone were associated with a higher proportion of below average QoL in physical OR 30.2, 95% CI 2.47, 370,

**Funding:** The author(s) received no specific funding for this work.

**Competing interests:** The authors have declared that no competing interests exist

psychological OR 9.54, 95% CI1.09,83.27 and social domains OR 5.94, 95% CI 1.25,28.34. In males, participants' sources of income from relatives were associated with a higher proportion of below average QoL in physical OR 3.6, 95% CI 2.01,6.44, psychological 30.2, 95% CI 2.47, 370, psychological OR 4.63 95% CI 2.56, 8.38, social domains OR 1.81, 95% CI 1.04, 3.16 and environmental domains OR 2.53 95% CI 1.44, 4.43 than those who had own income. Females engaged in income generation activities had better QoL in social and environmental domains than those with house duties, irrespective of their education or socioeconomic status. Males in retired life had the highest QoL in the social and environmental domains if they had better SES.

## Conclusions

The study's findings reveal that more than 50% of people had below-average QoL in each domain, which is significant. The study also highlighted that females living alone and males without their own income had the poorest QoL. On a positive note, it was found that females engaged in any income generation activities had better QoL in social and environmental domains. These results provide valuable insights for policymakers and healthcare professionals. They underscore the importance of implementing appropriate intervention programs to enhance the QoL of older adults, reiterating the urgency and necessity of their work to improve the health and well-being of older adults.

## Introduction

Every country in the world is experiencing growth in both the size and the proportion of older persons. It is projected that 1 in 6 people worldwide will be 60 years or over by 2030 [1,2]. Women have higher life expectancy than men, and they are susceptible to different disease patterns, which require different health-seeking behaviours than men [3]. For example, women need more care in residential environments, movement, connection, and emergency services than men. Men feel lonelier and more depressed than women [3,4]. With people's longer life expectancy, living arrangements also change and varies between genders. In 2020, approximately one-third of women aged 65 and older in the United States lived alone compared to 20% of older men [5]. In South Korea, the percentage of older adults living alone was 23.6% in 2017 (19.5% were men, 80.6% were women). The proportion of lonely deaths among older adults aged 60 years and older was reported to have increased significantly: 37.1% in 2017, 42.7% in 2019 and 47.5% in 2021[6]. Older men and women need different levels of community service. For example, in a study among 1023 Korean older adults, 66.5% of men vs. 76.5% of women needed community support for any injury or accident. Gender and health shape aging decisively, thereby impacting the quality of life (QoL) in old age, and that is relevant in recognizing societal developments as well as in formulating adequate strategies for facing the challenges of aging according to their need [7].

World Health Organization (WHO) recommends healthy aging, and quality of life (QoL) throughout life [8]. Older adults must be socially, economically and culturally active during aging. As such, an upsurge in research dedicated to examining the health and well-being of this specific group has been continued, with a particular focus on QoL [9–12]. There are several tools for measuring QoL, but most of the research on QoL has focused on the impact of chronic diseases, such as cancer, stroke and diabetes [13,14]. The WHO quality of life

(WHOQoL) was developed using cross-cultural, multinational studies on the concept of QoL at the population level and proposed four fundamental domains: physical health, psychological health, social relationships, and environmental health [15,16]. This questionnaire has been used for extensive population studies, including its use in Bangladesh [17–19]. Uddin and Islam [19] studied the psychometric properties of the WHOQoL-BREF 26-item questionnaire using Rasch analysis [20]. They proposed a modified version of 19-item tool, which was re-evaluated in a different sample and proposed the retention of the 19-item tool for its use in the general population [21].

Previous research has shown that factors such as age, gender, marital status, education, place of living, health status, employment, and socioeconomic status are associated with the QoL[22–25].

Bangladesh is a populated country with approximately 163 million people [26], with a steady increase in life expectancy: 65.05 years in 2000, 73.0 years in 2020, 73.82 years in 2024 and projected to be 79.24 years in 2050. One in every ten citizens will be 60 years old or older by 2025, which is expected to increase to one in every five by 2050 [27]. Several studies have reported health-related QoL in Bangladesh, with different diseases and conditions and specific populations [28–33]. For example, Amin et al. [28] assessed domain specific QoL and their determinants among 500 type 2 diabetes patients aged 16 and above using the WHOQoL-BREF [34]. However, none of the studies were conducted among older adults or investigated if there was a gender effect in associated factors with QoL domains. In a study among older adults aged 60 years or older in India, the QoL score was significantly lower than those in adults. To date, most research focusing on the QoL of older adults has been conducted in developed, high-income Western countries, including Canada and the USA [35,36]. However, due to the increasing trend in the aging population, the domain-specific QoL among older adults, their associated factors and whether there are any differences between genders warrant investigation. Therefore, this study aims to quantitatively examine the level of QoL among older adults in rural Bangladesh with a particular focus on gender difference and investigate relationships between their QoL and potential predictive factors, especially older adults' living arrangements and source of income.

## Materials and methods

### Study design, sample size and statistical power

This cross-sectional study was conducted between May and July 2017 in the Narail district of Bangladesh using a multi-stage cluster random sampling technique. Narail has an estimated population density of 722 people per km$^2$, which is comparable to the national rural population density of 873 people per km$^2$. Approximately 7% of residents of Narail are aged between 60 and 90 [37].

The current research was based on a fixed sample size of 1174 older adults (573 males and 574 females) aged 60–90 years. The study was designed to investigate psychological distress and QoL among rural people, which was previously described [38,39]. The required sample size is based on the study findings to show that the sample size was adequate for the current research. The current research indicates that the average quality of life (QoL) score in the psychological domain was 38.7 for males and 41.0 for females, with a pooled standard deviation of 13.9. The average value for the psychological domain was less than those of other domains. To calculate the sample size, the formula $n = (Z_{\alpha/2}+Z_{\beta})2 \ast 2 \ast \sigma^2 / d^2$ was used, where $Z_{\alpha/2}$ represents the critical value of the normal distribution at α/2 (e.g., for a confidence level of 95%, α is 0.05 and the critical value is 1.96), $Z_{\beta}$ represents the critical value of the normal distribution at β (e.g., for a power of 80%, β is 0.2 and the critical value is 0.84), σ2 is the population variance,

and d is the difference to detect. The given parameters produce a sample size of 572. Therefore, the current sample sizes of 573 males and 574 females were sufficient to detect a significant difference for all domains if the differences existed. For comprehensive results, the QoL was dichotomised based on below-average QoL. In the psychological domain, the below-average QoL in females was 273 (47.6%) and in males 244 (42.6%), which yielded a risk ratio (95% confidence interval) of 1.12 (0.98, 1.27). For each cell frequency for the exposed and unexposed group 100, if the difference in the proportion of outcome is at least 14% (e.g., 50% vs. 36%), it produces a significant risk ratio. For any cell frequency less than 100, the difference in outcome measures was required to be more than 14% to detect a significant risk ratio.

## Recruitment and methods of data collection

A multistage cluster random sampling technique was used for data collection. There are 13 rural unions in the rural area and nine wards under the urban city of Narail district. To better understand the administrative division, Bangladeshis are divided into eight main administrative divisions, each of which is divided into several districts, and thus, there are 64 districts or zila, including Narail. Each district comprises several Upazilas, each again divided into several rural unions and an urban pourassava. A rural union has several villages with 200–400 households, and an urban pourashava is subdivided into several wards, which again divide into several mahallas that comprise approximately 500 households. Narail is such a upazila. Three unions from a total of 13 rural unions and one ward from a total of nine urban wards of Narail were randomly selected at level 1. Two or three villages or mahallas from each union or ward were randomly chosen at level 2. Approximately 120 older adults from each village were interviewed. For data collection, first, the chief investigator at Swinburne University of Technology, in cooperation with the Organization for Rural Community Development (ORCD), a non-government organization in Bangladesh, recruited six data collectors. Second, all data collectors participated in an intensive two-day training program in Narail before they commenced the survey. The purpose of the training was to outline the study's rationale, procedures, and potential difficulties associated with data collection. Two interviewers were in one group for data collection, comprising three teams. Third, the data collectors contacted village leaders for their consent to collect data. Once they received the consent, they approached the prospective participants to explain the purpose of the study and to invite them to participate. Interviews were conducted upon their consent. One female was interviewed immediately after an interview of a male participant to maintain an approximately equal number of male and female participants. A pre-test of the questions in the final questionnaire was conducted by interviewing five lay people. Five women and men who were independent of the study team were interviewed using the questionnaire and were asked to assess comprehension, wording, and appropriateness. Five per cent of the entire data was entered twice by independent data entry operators to assure accuracy. Recruitment strategy and quality assurance in data collection were described in detail previously [38,39].

## Measuring QoL using modified WHOQoL-BREF as the dependent variable

Different instruments, both generic and disease-specific, can measure QoL. WHOQoL-BREF [34] with 26 items, is a generic instrument to measure subjective QoL and is a known and acceptable cross-cultural comparison instrument available in more than 40 countries. Of 26 items, two are about general health and overall QoL, seven are about physical health, six are about psychological conditions, three are about social values, and eight are about surrounding environmental conditions. Each item scores 1 to 5 on a Likert scale where 1 represents (very dissatisfied/very poor), and 5 represents (Very satisfied). WHOQoL-BREF of 26 items was

validated by Uddin and Islam [19,21] and has proposed 19-item validated tools to measure QoL in rural Bangladesh. The modified 19-item tool [21] has five items for the physical domain, four items for the psychological domain, three items unchanged for the social domain and five items for the environmental domain. The modified 19-item tool showed adequate internal consistency, reliability, unidimensionality and freedom from differential item functioning for sex and age.The original twenty-six items in the WHOQoL-BREF questionnaire, including domain names and item numbers is shown in S2 Data. Rasch analysis [20] converts categorical values into quantitative logit scores based on a person's ability and item difficulty with positive and negative values, a mean score of zero, and a standard deviation of 1. Then, the logit scores were converted on a 0 (worst QoL) to 100 (best QoL) score, 50 being the average QoL. This approach was proposed by Wright and Masters [40] in Rasch measurement theory. The study protocol, including methodology, can be found elsewhere [41].

The original 26 items and domain-specific items used in this research are shown in S1 Data.

### Independent variables

In this study, socio-demographic characteristics were independent variables. The socio-demographics considered in this study were age (i.e. age 60–69, 70–79 or aged 80+), gender (i.e. male or female), level of education–categorised into no schooling, primary to secondary school level of education (grade 1 to 10) and school secondary certificate (SSC) or above were collected. Since there is no taxable income for the village people and difficult to measure socio-economic status (SES) directly, a crude measure of SES was adapted according to Cheng et al. [42]. The question was asked whether, "Over the last twelve months, in terms of household food consumption, how would you classify your socioeconomic status?" the possible answers were: (i) insufficient funds for the whole year, (ii) insufficient funds for some of the time, (iii) neither deficit nor surplus (balance), and (iv) sufficient funds most of the time. Data on current occupation was house duties, engaged in income activities, retired and unable to work, marital status (married, widowed, never married or single), living status (living with family, living with relatives or non-relatives, living alone), and Sources of income (own income, support from family or children and government help or other sources).

### Ethics approval and consent to participate

We conducted the research following the tenets of the Declaration of Helsinki. Human Ethics Approval was received from the Swinburne University of Technology Human Ethics Committee (SHR Project 2015/065). We obtained written consent from the participants who were able to sign, and a fingerprint was obtained from those who were unable. In the case of fingerprint consent, the data collector provided a counter signature for the participants. Participants were informed of their right to withdraw from the study at any stage if they wished. Participants were informed that they had full rights to withdraw from the study at any stage if they wished and that their decision to participate or not would not influence their relationship with ORCD.

### Statistical analysis

Participants' characteristics, including age, level of education, occupation, and socioeconomic status, were presented using descriptive statistics. Data were presented as numbers and percentages as all variables were categorical or were categorized into a few categories, such as age and level of education. A generalized linear model was used to present adjusted means and standard error of QoL for four domains by different factors. The QoL percentage on the linear scale from zero to 100 was divided into two categories: a percentage less than 50 was below

average QoL, and 50 or above was equal or above average QoL. A binary logistic regression technique was used to estimate the odds ratio (OR) with 95% confidence intervals (CI) for each factor for below average QoL. The reference categories included age 60–69 years for the age group, no education level for the level of education, married for marital status, engaging in any income generation activities for occupation, living with family for living status, and own income for sources of income variables. The reference groups are shown in the results section. When an interaction between variables and gender was sought, the confounding variables were age, education, occupation, marital status, living status, and sources of income for model adjustment for the total sample, and to obtain the interaction p values. The variable gender was removed when stratification analyses for gender were obtained. RUMM 2030 [43] was used for Rasch analysis to compute the combined score from each domain. Statistical software SPSS (SPSS Inc, version 27) was used for descriptive data, GLM and logistic regression. Any results were statistically significant with $p \leq 0.001$ for less than or equal to 0.1% significance level. Other than less than or equal to 0.1% significance level, the actual p values were presented. The odds ratio was significant if the lower and upper limit of the confidence interval was less than 1.0 for an odds ratio less than 1.0 and greater than or equal to one for an odds ratio greater than 1.0.

## Results

Table 1 describes the socio-demographic characteristics, including the participants' living status and sources of income by gender (male vs female). Although the male and female participants were equal, female participants were younger, and more than half of the females had no education compared to one-third had no education among male participants. None of the male participants were homemakers. Of females, only 1.6% were engaged in any income activities, whereas this percentage was almost half among males. Over half of the females were widowed, and only one-tenth of the males were widowed. Over one-third of males had their income, but only 4.5% of females had it, and more than 90% were dependent on their children or family income.

### Domain specific quality of life

The domain-specific QoL expressed in percentage, the mean (standard error) and minimum-maximum values were physical domain 48.9 (0.41), and 7–86, psychological domain 38.9 (0.51) and 4–71, social domain 50.5 (0.49) and 8–92, and environmental domain 47.8 (0.37) and 6–91 in the total sample with significant gender difference in physical, psychological and environmental domains. Higher socioeconomic status was significantly associated with higher endorsement in all QoL domains for males in the total sample. There were also significant interactions between different factors and gender in the association of the QoL domains (Table 2). In the case of the physical domain, the significant interaction was between the source of income and gender (<0.001) (Fig 1). In the case of the psychological domain, the significant interactions were education (p = 0.002), occupation (p = 0.006), and socioeconomic status (p = 0.003) with gender (Figs 2–4), respectively. For example, irrespective of education level, females' QoL was low, whereas males with having SSC or above, being retired, engaged in any income activities, or having sufficient funds most of the time had significantly higher QoL in the psychological domain. In the case of the social domain, the significant interactions were age group (p = 0.02), education (p = 0.003), occupation (p = 0.02), socioeconomic status (p<0.001), marital status (p<0.001) and income source (p = 0.005) with gender (Figs 5–10), respectively. For example, females aged 60–79 years had a mean QoL above 50% in the social domain, which drops to 41% after age 80. However, males' social domain QoL was similar

**Table 1. Socio-demographic characteristics of the elderly population in the Narail district of Bangladesh by gender.**

| | Total, N = 1147 | | Female, N = 574 | | Male, N = 573 | | p* |
|---|---|---|---|---|---|---|---|
| | N | % | N | % | n | % | |
| Age group | | | | | | | <0.001 |
| 60–69 | 729 | 63.6 | 399 | 69.5 | 330 | 57.6 | |
| 70–79 years | 315 | 27.5 | 131 | 22.8 | 184 | 32.1 | |
| 80+ | 103 | 9.0 | 44 | 7.7 | 59 | 10.3 | |
| Education level | | | | | | | <0.001 |
| No Education | 500 | 43.6 | 298 | 51.9 | 202 | 35.3 | |
| Primary to below school secondary certificate (SSC) (year 1–10) | 525 | 45.8 | 247 | 43.0 | 278 | 48.5 | |
| School Secondary Certificate and above | 122 | 10.6 | 29 | 5.1 | 93 | 16.2 | |
| Occupation | | | | | | | <0.001 |
| House duties | 341 | 29.7 | 341 | 59.4 | 0 | 0.0 | |
| Engaged in income activities | 271 | 23.6 | 9 | 1.6 | 262 | 45.7 | |
| Retired | 101 | 8.8 | 22 | 3.8 | 79 | 13.8 | |
| Unable to work | 434 | 37.8 | 202 | 35.2 | 232 | 40.5 | |
| Socioeconomic status | | | | | | | 0.01 |
| Insufficient funds for the whole year, | 195 | 17.0 | 116 | 20.2 | 79 | 13.8 | |
| Insufficient funds for some of the time | 381 | 33.2 | 194 | 33.8 | 187 | 32.6 | |
| Neither deficit nor surplus (balance), | 455 | 39.7 | 208 | 36.2 | 247 | 43.1 | |
| Sufficient funds most of the time | 116 | 10.1 | 56 | 9.8 | 60 | 10.5 | |
| Marital status | | | | | | | <0.001 |
| Married | 774 | 67.5 | 269 | 46.9 | 505 | 88.1 | |
| Widowed | 367 | 32.0 | 301 | 52.4 | 66 | 11.5 | |
| Unmarried or never married | 6 | 0.5 | 4 | 0.7 | 2 | 0.3 | |
| Living status | | | | | | | 0.004 |
| With family | 1126 | 98.2 | 556 | 96.9 | 570 | 99.5 | |
| With other relatives or non-relatives | 7 | 0.6 | 6 | 1.0 | 1 | 0.2 | |
| Living alone | 14 | 1.2 | 12 | 2.1 | 2 | 0.3 | |
| Sources of income | | | | | | | <0.001 |
| Own income | 240 | 20.9 | 26 | 4.5 | 214 | 37.3 | |
| Support from family or children | 873 | 76.1 | 533 | 92.9 | 340 | 59.3 | |
| Government help or other sources | 34 | 3.0 | 15 | 2.6 | 19 | 3.3 | |

*p ≤0.001 for less than or equal to 0.1% significance level; for the rest the actual p values are shown.

between ages 60 to 80 years. Females' QoL in the social domain was the minimum even if they had an education of SSC or above, but males had the highest average QoL with SSC or above education. In the case of the environmental domain, the significant interactions were education ($p < 0.001$), occupation ($p = 0.003$), socioeconomic status ($p < 0.001$), and income source ($p = 0.005$) with gender (Figs 11–14), respectively. Both males and females who were unable to work or engaged in any income activities had similar QoL in the environmental domain, but males who had SSC or above education had significantly higher QoL than females.

Table 3 shows the proportion of below-average QoL in the physical domain and psychological domain and Table 4 shows the social and environmental domains. The proportion of below-average QoL in the physical domain was 45.1, the psychological domain 70.5, the social domain 32.7 and the environmental domain 52.6. The proportion of below-average QoL for

**Table 2. Factors associated with quality of life (four domains: Physical, psychological, social and environmental) measured in 0–100 scale using WHOQoL-BREF questionnaire of 1147 elderly people, and by gender.**

| | Number (Male) | Physical domain | | | Psychological domain | | | Social domain | | | Environmental domain | | |
|---|---|---|---|---|---|---|---|---|---|---|---|---|---|
| | | Total | Female | Male | Total | Female | Male | Total | Female | Male | Total | Female | Male |
| | | Mean (SE)* | Mean (SE)* | Mean (SE)* | Mean (SE)* | Mean (SE)* | Mean (SE)* | Mean (SE)* | Mean (SE)* | Mean (SE) | Mean (SE)* | Mean (SE)* | Mean (SE)* |
| Total | 1147 (573) | 48.9 (0.41) | 47.9 (0.56) | 49.9 (0.60)+ | 39.8 (0.52) | 38.7 (0.60) | 41.0 (0.60)+ | 50.5 (0.49) | 49.9 (0.72) | 51.2 (0.66)+ | 47.8 (0.37) | 46.4 (0.52) | 49.2 (0.53)+ |
| Age group† | | 0.35 | | | 0.12 | | | 0.02 | | | 0.20 | | |
| 60–69 | 729 (300) | 52(0.5) | 51(0.6) | 54(0.7) | 42(0.5) | 40(0.7) | 44(0.8) | 51(0.6) | 50(0.8) | 52(0.9) | 48(0.5) | 47(0.6) | 50(0.7) |
| 70–79 years | 315 (184) | 44(0.7) | 42(1.1) | 45(1.0) | 36(0.7) | 36(1) | 36(1.0) | 52(0.9) | 53(1.4) | 51(1.1) | 47(0.7) | 47(1.1) | 48(0.9) |
| 80+ | 103 (59) | 40(1.5) | 37(2.0) | 43(2.1) | 36(1.4) | 33(2.1) | 39(1.7) | 47(1.8) | 43(3.1) | 51(2.2) | 47(1.3) | 44(2) | 50(1.7) |
| P** | | <0.001 | <0.001 | <0.001 | <0.001 | <0.001 | <0.001 | 0.07 | 0.003 | 0.74 | 0.66 | 0.29 | 0.27 |
| Education† | | 0.65 | | | 0.002 | | | 0.003 | | | <0.001 | | |
| No Education | 500 (202) | 49 (1.1) | 47 (1.4) | 49 (1.5) | 38 (1.1) | 38 (1.5) | 37 (1.6) | 48 (1.5) | 49 (2.0) | 48 (1.8) | 46 (1) | 46 (1.3) | 48 (1.3) |
| Below school secondary certificate | 525 (278) | 50 (1) | 47 (1.5) | 51 (1.3) | 39 (1.1) | 35 (1.6) | 43 (1.4) | 50 (1.4) | 45 (2.2) | 55 (1.6) | 46 (1) | 42 (1.4) | 51 (1.1) |
| School secondary certificate and above | 122 (93) | 51 (1.7) | 53 (2.5) | 51 (2) | 39 (1.8) | 42 (2.7) | 42 (2.1) | 48 (2.3) | 47 (3.7) | 51 (2.4) | 48 (1.5) | 48 (2.4) | 52 (1.7) |
| P** | | 0.42 | 0.35 | 0.63 | 0.47 | 0.63 | 0.31 | 0.25 | 0.90 | 0.05 | 0.14 | 0.50 | 0.45 |
| Occupation† | | 0.12 | | | 0.006 | | | 0.02 | | | 0.003 | | |
| House duties | 341 (0) | 50 (1.3) | 50 (1.3) | | 40 (1.4) | 40 (1.4) | | 48 (1.8) | 48 (1.9) | | 44 (1.2) | 44 (1.2) | |
| Engaged in income activities | 271 (262) | 53 (1.4) | 48 (3.7) | 53 (1.5) | 40 (1.5) | 37 (4.1) | 41 (1.5) | 49 (2.0) | 44 (5.4) | 51 (1.8) | 45 (1.3) | 43 (3.5) | 46 (1.3) |
| Retired | 101 (79) | 49 (1.6) | 50 (2.7) | 49 (1.9) | 40 (1.7) | 36 (3) | 42 (1.9) | 49 (2.2) | 46 (4) | 52 (2.2) | 50 (1.5) | 45 (2.6) | 53 (1.6) |
| Unable to work | 434 (187) | 47 (1.3) | 43 (1.6) | 48 (1.5) | 35 (1.4) | 34 (1.8) | 37 (1.5) | 48 (1.8) | 47 (2.4) | 51 (1.8) | 47 (1.2) | 47 (1.5) | 50 (1.3) |
| P** | | 0.90 | 0.77 | 0.62 | 0.33 | 0.83 | 0.25 | 0.63 | 0.35 | 0.99 | 0.78 | 0.43 | 0.03 |
| Socioeconomic status | | 0.10 | | | 0.003 | | | <0.001 | | | <0.001 | | |
| Insufficient funds for the whole year, | 195 (79) | 43 (1.4) | 43 (1.7) | 43 (2) | 32 (1.5) | 33 (1.9) | 32 (2.1) | 43 (1.9) | 45 (2.5) | 42 (2.5) | 36 (1.3) | 36 (1.6) | 38 (1.8) |
| Insufficient funds for some of the time | 381 (187) | 48 (1.4) | 48 (1.9) | 46 (1.8) | 34 (1.5) | 35 (2.1) | 33 (1.9) | 50 (1.9) | 52 (2.8) | 50 (2.2) | 45 (1.3) | 46 (1.8) | 45 (1.6) |
| Neither deficit nor surplus (balance), | 455 (247) | 57 (1.2) | 53 (1.7) | 57 (1.6) | 46 (1.3) | 44 (1.8) | 46 (1.6) | 52 (1.6) | 50 (2.4) | 53 (1.9) | 53 (1.1) | 51 (1.6) | 54 (1.4) |
| Sufficient funds most of the time | 116 (60) | 47 (1.6) | 45 (2.2) | 49 (2.1) | 43 (1.7) | 39 (2.4) | 46 (2.2) | 50 (2.2) | 41 (3.3) | 59 (2.5) | 53 (1.5) | 48 (2.1) | 61 (1.8) |
| P** | | <0.001 | 0.39 | 0.01 | <0.001 | 0.85 | <0.001 | 0.52 | 0.31 | 0.05 | <0.001 | 0.02 | <0.001 |
| Marital status† | | 0.11 | | | 0.06 | | | <0.001 | | | 0.19 | | |
| Married | 774 (505) | 51 (0.8) | 47 (1.4) | 51 (1) | 40 (0.9) | 40 (1.6) | 41 (1) | 49 (1.1) | 49 (2.1) | 49 (1.2) | 48 (0.8) | 48 (1.4) | 49 (0.9) |
| Widowed | 367 (66) | 47 (1.2) | 47 (1.2) | 46 (2.3) | 36 (1.3) | 35 (1.4) | 38 (2.4) | 48 (1.6) | 46 (1.8) | 57 (2.7) | 44 (1.1) | 42 (1.2) | 53 (2) |
| Unmarried or never married | 6 (2) | 55 (4.7) | 57 (5.6) | 52 (8.3) | 38 (5) | 40 (6.1) | 33 (8.7) | 55 (6.4) | 50 (8.1) | 65 (10) | 47 (4.4) | 43 (5.3) | 55 (7.2) |
| P** | | 0.62 | 0.23 | 0.67 | 0.64 | 0.36 | 0.82 | 0.81 | 0.56 | 0.21 | 0.44 | 0.53 | 0.98 |
| Living status† | | 0.29 | | | 0.87 | | | 0.35 | | | 0.39 | | |
| With family | 1126 (570) | 51 (0.7) | 49 (1) | 50 (0.9) | 40 (0.7) | 39 (1.1) | 40 (1) | 50 (1) | 49 (1.4) | 52 (1.1) | 48 (0.7) | 47 (0.9) | 50 (0.8) |
| With relatives or non-relatives | 7 (1) | 35 (4.3) | 37 (4.5) | 20 (11.8) | 28 (4.6) | 28 (5) | 24 (12.2) | 36 (5.9) | 39 (6.6) | 16 (14.2) | 33 (4) | 35 (4.3) | 21 (10.3) |
| Living alone | 14 (2) | 44 (3.5) | 43 (3.8) | 50 (8.3) | 30 (3.7) | 29 (4.2) | 35 (8.7) | 40 (4.8) | 39 (5.6) | 41 (10) | 32 (3.3) | 31 (3.6) | 36 (7.3) |
| P** | | 0.24 | 0.46 | 0.07 | 0.20 | 0.23 | 0.87 | 0.90 | 0.68 | 0.86 | 0.49 | 0.48 | 0.26 |
| Sources of income† | | <0.001 | | | 0.06 | | | 0.005 | | | 0.02 | | |

(*Continued*)

**Table 2.** (Continued)

| | | Physical domain | | | Psychological domain | | | Social domain | | | Environmental domain | | |
|---|---|---|---|---|---|---|---|---|---|---|---|---|---|
| | | Total | Female | Male | Total | Female | Male | Total | Female | Male | Total | Female | Male |
| | Number (Male) | Mean (SE)* | Mean (SE)* | Mean (SE)* | Mean (SE)* | Mean (SE)* | Mean (SE)* | Mean (SE)* | Mean (SE)* | Mean (SE) | Mean (SE)* | Mean (SE)* | Mean (SE)* |
| Own income | 240 (214) | 56 (1.5) | 53 (2.4) | 58 (1.7) | 43 (1.6) | 43 (2.7) | 43 (1.8) | 47 (2) | 47 (3.6) | 50 (2.1) | 47 (1.4) | 44 (2.3) | 51 (1.5) |
| Support from family or children | 873 (340) | 47 (0.8) | 47 (1.0) | 45 (1.1) | 38 (0.8) | 37 (1.1) | 38 (1.2) | 49 (1.1) | 46 (1.5) | 53 (1.3) | 47 (0.7) | 45 (0.9) | 51 (1) |
| Government help or other sources | 34 (19) | 48 (2.1) | 44 (2.9) | 53 (2.9) | 35 (2.3) | 33 (3.2) | 40 (3.1) | 49 (2.9) | 53 (4.2) | 46 (3.6) | 44 (2) | 45 (2.7) | 44 (2.6) |
| P** | | 0.29 | 0.42 | <0.001 | 0.29 | 0.10 | 0.02 | 0.40 | 0.21 | 0.66 | 0.49 | 0.96 | 0.02 |

*Means (standard error) adjusted for age, education, occupation, marital status, living status and sources of income for the toral sample and to obtain the interaction p values. Gender was excluded from the adjusted covariates for stratification for gender analysis; + significant difference between gender in the total sample

p** is for factors associated with domain specific QoL in the total sample and within genders; † is for interaction between various factors with gender in association of domain specific QoL in the total sample.

females vs. males was: physical domain 47.6% vs. 42.6%, psychological domain 74.4% vs. 66.7%, social domain 34.8% vs. 30.1%, and the environmental domain was 56.1% vs. 49.0%. Among females, the proportion of below average QoL in physical domain odds ratio (OR) 6.04, 95% confidence interval (CI)): 3.63–10.06 and psychological domain OR 3.05, 95% CI: 1.73–5.38 was higher among those who were unable to work compared to those who had house duties. However, in the case of the social domain, OR 3.86, 95% CI:1.46–10.22 was higher among retired participants compared to those who had house duties occupation. Higher SES was associated with a lower proportion of below-average QoL in psychological and environmental domains for both genders. In females, the proportion of below average QoL in the social domain was higher among widowed OR 1.45, 95% CI: 1.0–2.22 and in males, the proportion was lower OR 0.33, 95% CI: 0.16, 0.67. In females, participants living alone were associated with a higher proportion of below average QoL in physical OR 30.2, 95% CI: 2.47–370, psychological OR 9.54, 95% CI: 1.09–83.27 and social domains OR 5.94, 95% CI: 1.25–28.34. In males, participants' sources of income from relatives were associated with a higher proportion of below average QoL in physical OR 3.60, 95% CI: 2.01–6.44, psychological OR

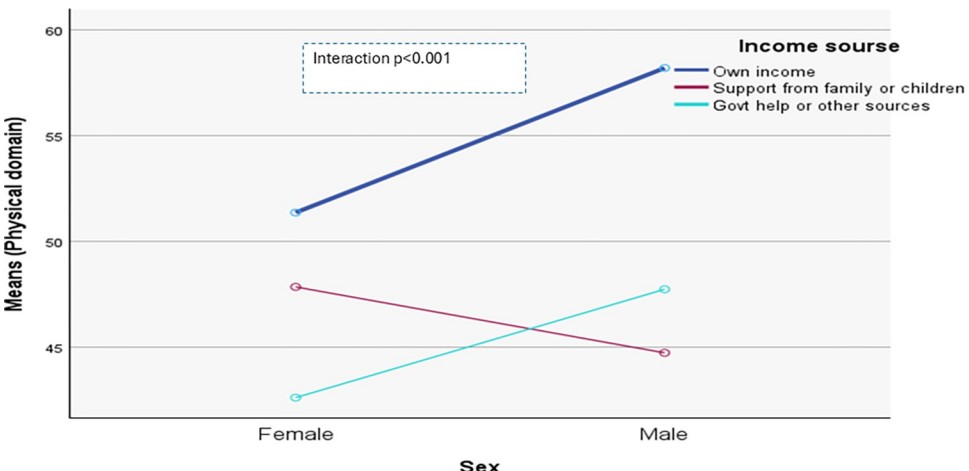

**Fig 1. Interaction between income source and gender in the association of QoL of the physical domain.**

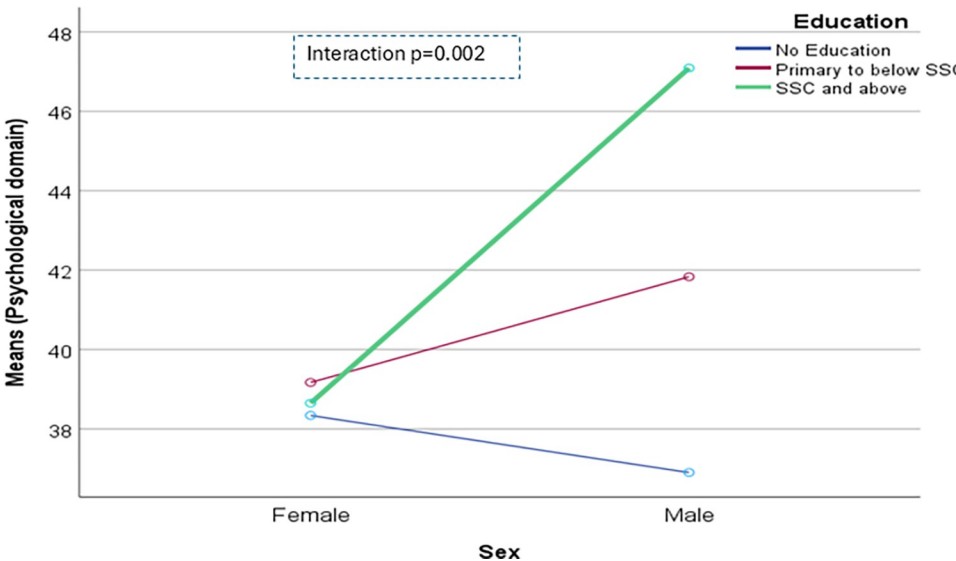

**Fig 2. Interaction between education and gender in the association of QoL of the psychological domain.**

4.63, 95% CI: 2.56–8.38, social OR 1.81, 95% CI: 1.04–3.16 and environmental domains OR 2.53, 95% CI: 1.44–4.43 than those who had own income. Such associations were not significant for females.

## Discussion

In this study, we report the mean QoL, the proportion of below-average QoL and their associated factors, including living status and sources of income in a large sample of older adults aged 60–90 years expected to have minimum or no income in Bangladesh by gender. The significant findings from this study include: (1) overall, there are significant interaction between

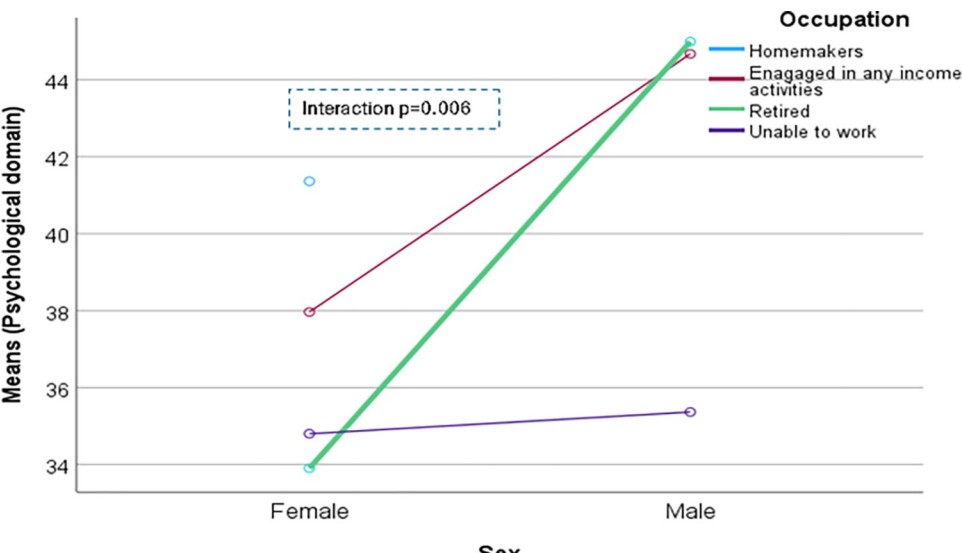

**Fig 3. Interaction between occupation and gender in the association of QoL of the psychological domain.**

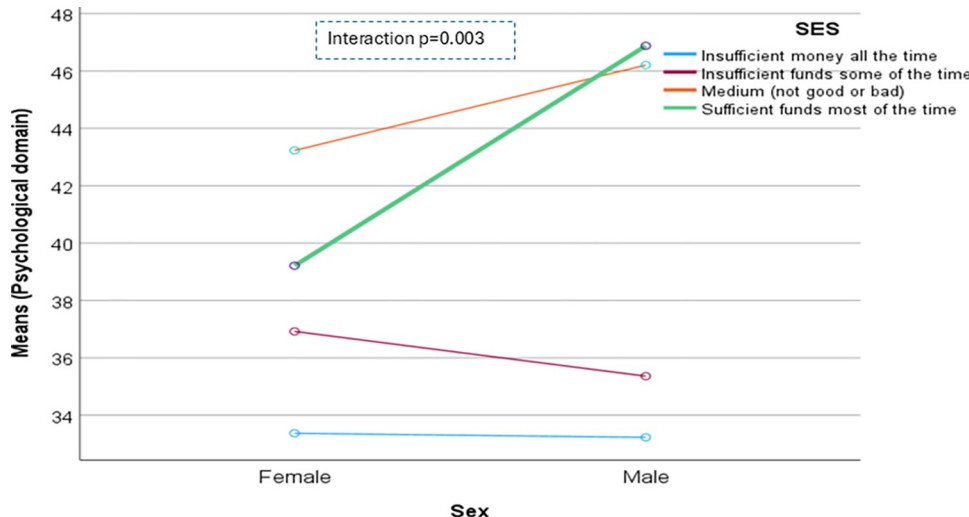

**Fig 4. Interaction between socioeconomic status (SES) and gender in the association of QoL of the psychological domain.**

various factors and gender in the association of QoL of all domains and the females had poorer QoL than males, (2) QoL was below average in all QoL measuring domains, and mean QoL in psychological domain had the most minimum, (3) People who were unable to work had the poorest QoL in physical domain, (4) females who lived alone had significantly poorer QoL in physical, psychological and social domains, (5) males who did not have income and were dependent on family income had poorer QoL in psychological, social and environmental domains, (6) retired males had the highest QoL in social domain if they had sufficient funds or had SSC or above education level in contrast retired males had the poorest QoL in social domain if they had insufficient funds all the time, (7) homemakers who had SSC or above education had the poorest QoL in environmental domain, and (8) females with low SES had higher QoL in social domain if they were engaged in any income generation activities.

Although the proportion of males and females is equal in the current study, their socioeconomic characteristics were different. For example, females' education level was lower, less engaged in income activities and more were widowed than males. The higher proportion of widows in females includes the fact that women have a longer life expectancy, marry men older than themselves, and are more likely to remain unmarried after their spouse dies.

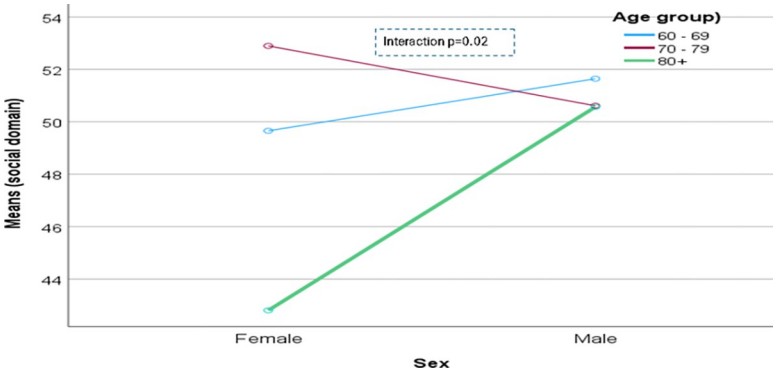

**Fig 5. Interaction between age group and gender in the association of QoL of the social domain.**

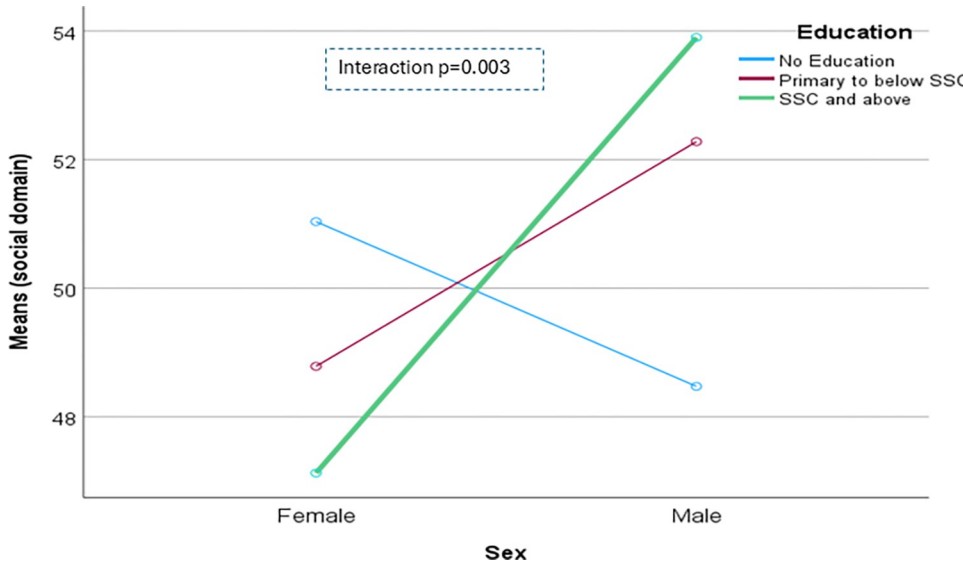

**Fig 6. Interaction between education and gender in the association of QoL of the social domain.**

Consequently, in every nation, a significantly higher proportion of women than men are widowed [3,44]. This difference between genders in marital status, living status and other socio-economic factors associated with QoL domains shows variations. The current study reported that the domain-specific mean QoL was less than 50%, inconsistent with previous studies. It was higher than those reported in some studies [23,28] and lower than others [22,32,45]. Thadathil1[23] studied rural people in India, and the current study was in a rural district in Bangladesh, and the age distribution was the same. The higher mean QoL in our study than the Indian population can be due to environmental and cultural variations.

Lodhi [22] conducted a study among 2063 general people aged 16 to 90 and reported significantly higher QoL in all domains. Although many characteristics of the participants in the two studies are similar, the reason for this difference is unknown. However, this can be because

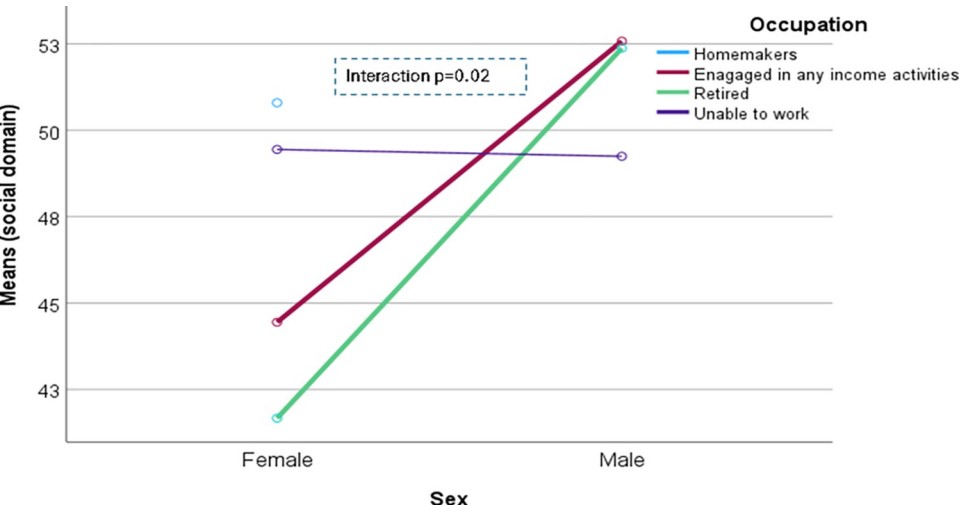

**Fig 7. Interaction between occupation and gender in the association of QoL of the social domain.**

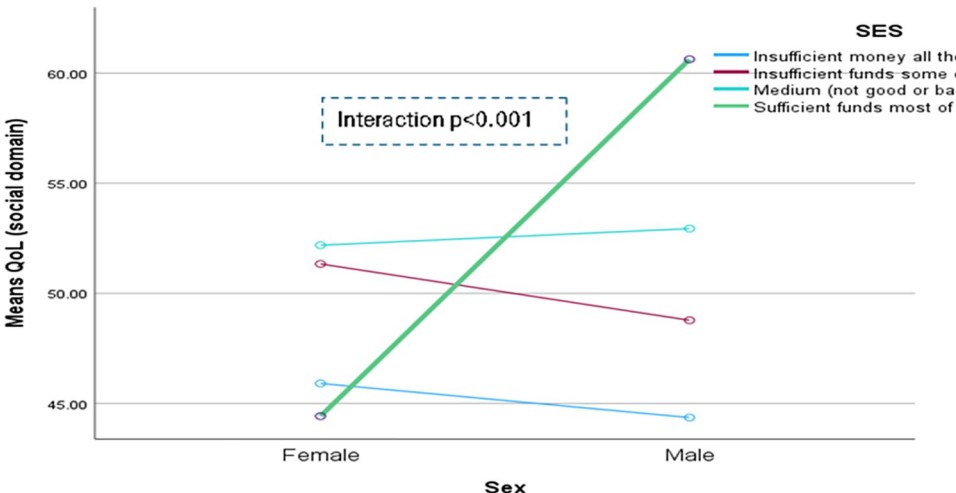

**Fig 8. Interaction between socioeconomic status (SES) and gender in the association of QoL of the social domain.**

of different countries, and inherently, the people in Pakistan could have better QoL. In the current study, higher QoL scores in males, younger people, and engaged in income-generation activities are consistent with previous studies [22,23,28,29,32,33,46], but not consistent with the level of education [22,23,28,33]. The current study severely affected the psychological domain, as it was found among older adults [23,28]. Similar to other studies, the current research shows that psychological feelings become more affected as age advances [22,23,33].

Amin et al. [28] reported a QoL score of 37.02 in the psychological domain in people with no education compared to 48.75 in people who had graduated. Compared to this finding, the current study reports a mean QoL score of 38.0 in people with no education and 39.0 in people with SSC or above education in the total sample. There was a significant interaction between

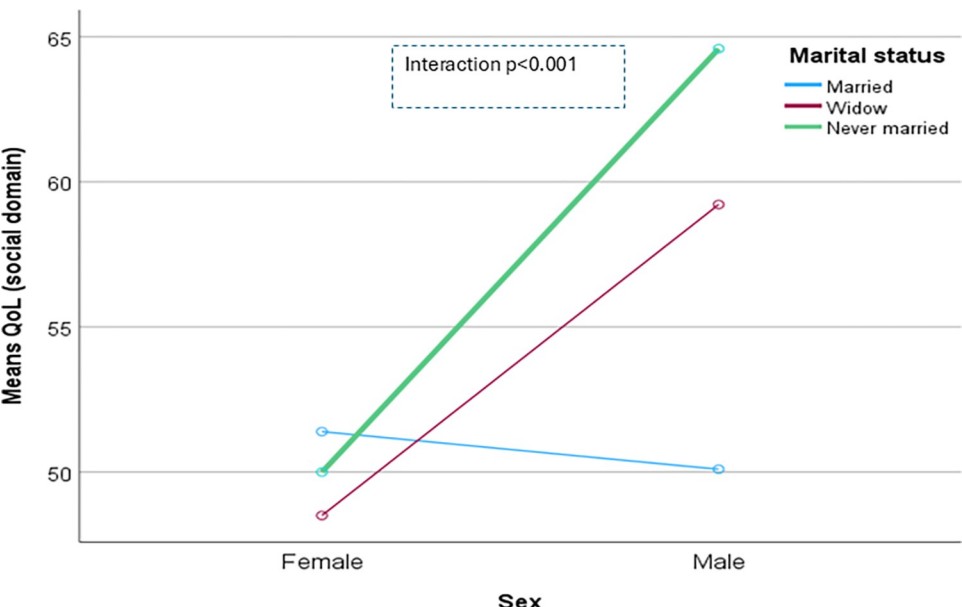

**Fig 9. Interaction between marital status and gender in the association of QoL of the social domain.**

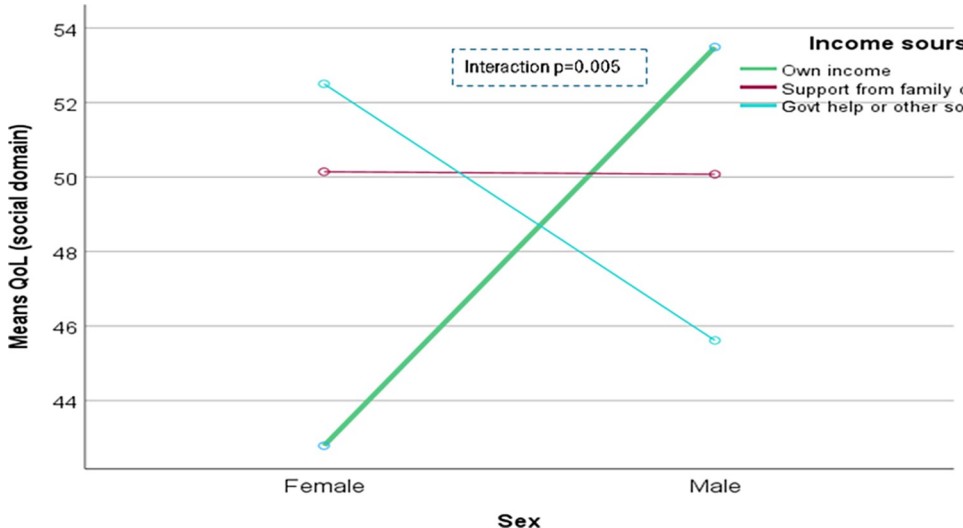

**Fig 10. Interaction between income source and gender in the association of QoL of the social domain.**

education and gender in the psychological domain; males who had SSC or above education were one-quarter likely to have a below average level of QoL in the psychological domain compared to those who had no education. Compared to people who were engaged in any income generation activities, males who were unable to work were more likely to have a higher proportion of below-average QoL physically, psychologically, and socially. Among females, the associations were also in the same direction. The association of higher incomes and engagement in income-generation activities with higher QoL in all domains, in both genders, is consistent with previous studies [23,28,33].

In the case of the environmental domain, retired females were more likely to have below-average QoL. The interaction between occupation and education in the environmental domain

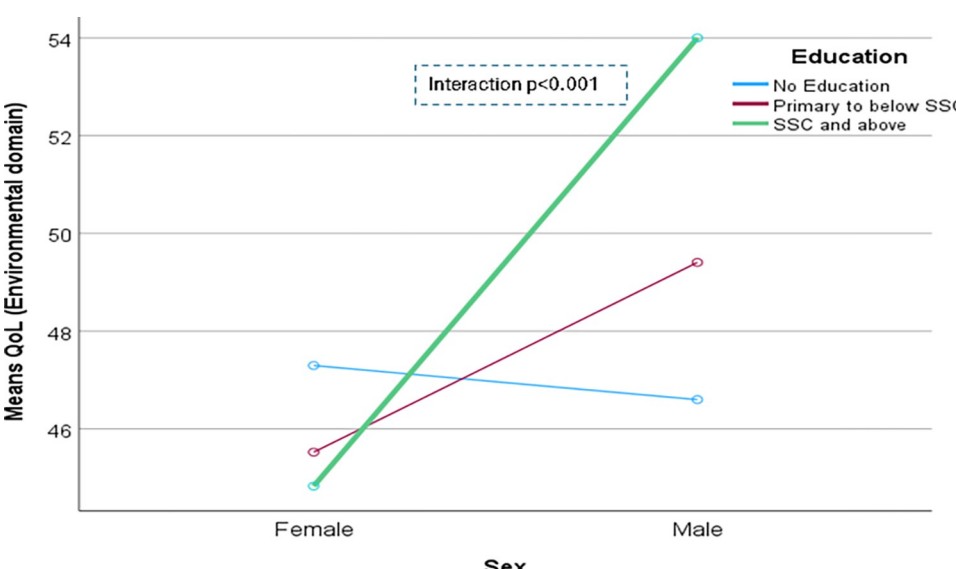

**Fig 11. Interaction between education and gender in the association of QoL of the environmental domain.**

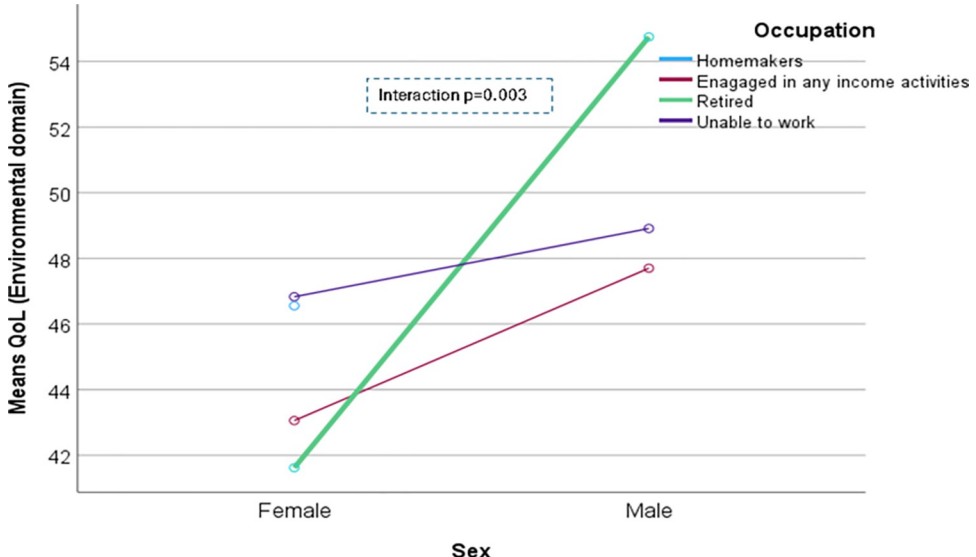

**Fig 12. Interaction between occupation and gender in the association of QoL of the environmental domain.**

revealed that QoL was significantly higher in those who were retired and had an education level of SSC or above in both genders. Still, irrespective of the level of education, males who were not engaged in income generation activities had significantly poorer QoL in the environmental domain. Retirees reported low self-esteem and decreased contact with friends and colleagues due to retirement [47]. Retirees feel devalued, unproductive, unengaged in social and work activities, transportation and leisure and miss the social interaction provided by work, leading to a great emotional void. The effect is more severe in disadvantaged groups, such as people with low levels of education [48]. The retirement process influences emotional and social dimensions and impairs the functionality of retired people. It can lead to physiological, psycho-emotional, and socioeconomic vulnerability, in addition to unexpected behaviour, psychopathologies, and new and inadequate attitudes [49,50]. Among males, irrespective of

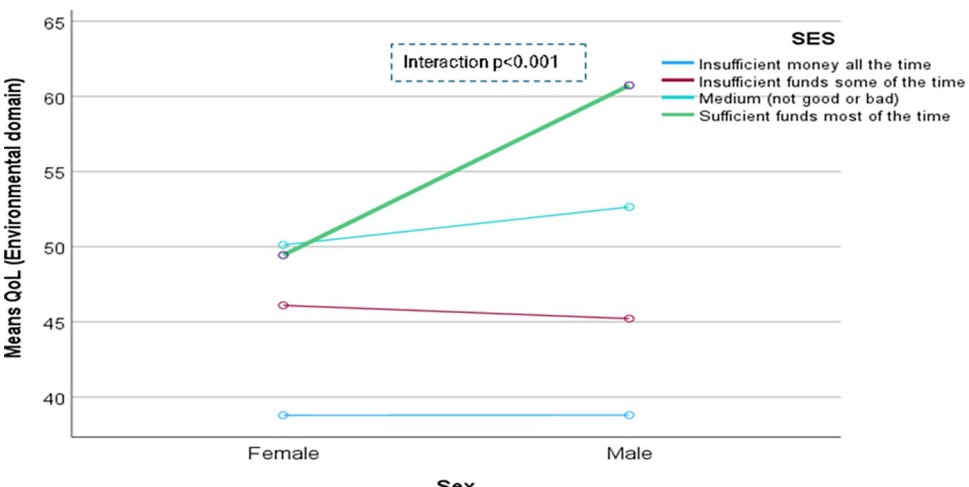

**Fig 13. Interaction between socioeconomic status (SES) and gender in the association of QoL of the environmental domain.**

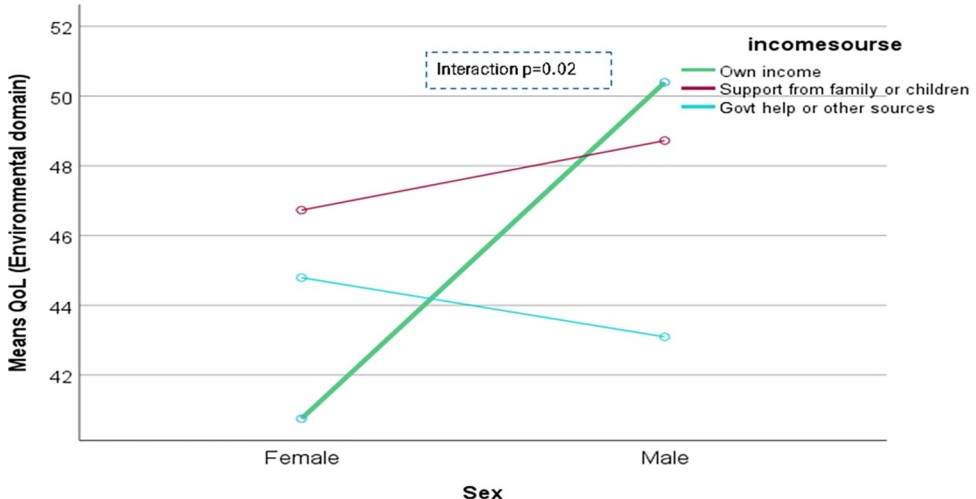

**Fig 14. Interaction between income source and gender in the association of QoL of the environmental domain.**

occupation, QoL in the social domain was significantly high if they had sufficient funds most of the time. However, among females, sufficient funds did not contribute to QoL in the social domain; instead, those who had insufficient funds but were engaged in income-generation activities had better QoL. In many developing countries worldwide, including Bangladesh, financial access or income generation activities, particularly among the poor and disadvantaged groups, have become a key weapon for eliminating poverty and improving QoL [51]. The lack of association of higher socioeconomic status with better QoL among females can be explained by the social and cultural fact that in many Asian countries, including Bangladesh, females do not enjoy much financial independence even if the family income is higher. In the current study, only males with no income or whose sources were either family or government support were associated with a higher proportion of below-average QoL in all domains. This finding signifies that in Bangladesh, men are more likely than women to participate in the labour force (81.3% vs. 42.5%), which implies that men are more vulnerable if they need to depend on someone else income than women who are socially acceptable to rely on family income [52].

In the current study, most people lived with their families. This result supports the coresidence of older adults living with children or grandchildren in developing nations [44]. The study reports that living alone or with relatives was associated with a higher proportion of below-average QoL in psychological, social and environmental domains. Such associations are consistent with the fact that family is known to be an essential source of social contact in older years [53]. Due to increased psychological distress and other comorbid conditions in the aging population, QoL decreases in social and environmental domains [54].

Bangladesh has seen a spike in older adults over the years due to the rapid increase in life expectancy. Population ageing presents social, economic and cultural challenges to individuals, families, societies and the global community [55]. Although traditionally older people reside with their children and grandchildren, these traditional family care practices are no longer sustainable due to dynamic demographic and social changes that have transformed family structures and concepts, as well as vast migration from rural areas to cities and abroad. Thus, the trend of the number of people living alone is increasing. As a result, older adults, especially women and widows, economically poor individuals, and without engagement in any income generation activities have become more vulnerable and often lack adequate care and support

**Table 3. Association of socio-demographic factors with quality-of-life domains (physical, psychological, social and environmental) of 1147 elderly people in a rural district Narail in Bangladesh.**

| | Female, N = 574 | Male, N = 573 | Physical domain, below average (<50%) QoL, N = 517 (45%) | | | | Psychological domain, below average (<50%) QoL, N = 809 (70.5%) | | | |
|---|---|---|---|---|---|---|---|---|---|---|
| | | | Female, 273 (47.6%) | | Male, 244 (42.6%) | | Female, 427 (74.4%) | | Male, 382 (66.7%) | |
| **Age group** | N | N | n(%) | OR (95% CI)* | n(%) | OR (95% CI)* | n(%) | OR (95% CI)* | n(%) | OR (95% CI)* |
| 60–69 | 399 | 330 | 153 (38.3) | 1.0 (ref.) | 101 (30.6) | 1.0 (ref.) | 280 (70.2) | 1.0 (ref.) | 189 (57.3) | |
| 70–79 years | 131 | 184 | 86 (65.6) | 1.88 (1.14,3.1) | 108 (58.7) | 1.78 (1.13,2.81) | 110 (84.0) | 1.87 (1.03,3.39) | 146 (79.3) | 1.9 (1.16,3.12) |
| 80+ | 44 | 59 | 34 (77.3) | 3.07 (1.3,7.24) | 35 (59.3) | 1.81 (0.92,3.59) | 37 (84.1) | 1.72 (0.67,4.42) | 47 (79.7) | 2.01 (0.92,4.41) |
| **Education level** | | | | | | | | | | |
| No Education | 298 | 202 | 166 (55.7) | 1.0 (ref.) | 117 (57.9) | 1.0 (ref.) | 220 (73.8) | 1.0 (ref.) | 156 (77.2) | 1.0(ref) |
| Below school secondary certificate | 247 | 278 | 98 (39.7) | 1.08 (0.7,1.66) | 104 (37.4) | 0.73 (0.47,1.14) | 188 (76.1) | 1.13 (0.77,1.67) | 183 (65.8) | 0.57 (0.38,0.86) |
| School secondary certificate and above | 29 | 93 | 9 (31.0) | 0.77 (0.29,2.04) | 23 (24.7) | 0.79 (0.38,1.65) | 19 (65.5) | 0.67 (0.30,1.51) | 43 (46.2) | 0.25 (0.15,0.43) |
| **Occupation** | | | | | | | | | | |
| Engaged in income activities | 9 | 262 | 4 (44.4) | 1.0 (ref.) | 58 (22.1) | 1.0 (ref.) | 6(66.7) | 1.0 (ref.) | 150 (57.3) | 1.0 (ref.) |
| House duties | 341 | 0 | 112 (32.8) | 0.61 (0.16,2.32) | – | – | 234 (68.6) | 1.09 (0.27,4.45) | – | – |
| Retired | 22 | 79 | 11 (50.0) | 1.25 (0.26,5.94) | 30 (38.0) | 2.15(1.26,3.69) | 19 (86.4) | 3.17 (0.5,20.04) | 43 (54.4) | 0.89 (0.54,1.48) |
| Unable to work | 202 | 232 | 146 (72.3) | 3.26 (0.84,12.6) | 156 (67.2) | 7.22 (4.84,10.77) | 168 (83.2) | 2.47 (0.59,10.37) | 189 (81.5) | 3.28 (2.17,4.95) |
| **Socioeconomic status** | | | | | | | | | | |
| Insufficient funds for the whole year | 116 | 79 | 69 (59.5) | 1.0 (ref.) | 45 (57) | 1.0 (ref.) | 103 (88.8) | 1.0 (ref.) | 71 (89.9) | |
| Insufficient funds for some of the time | 194 | 187 | 101 (52.1) | 0.53 (0.3,0.92) | 102 (54.5) | 0.56 (0.29,1.05) | 153 (78.9) | 0.35 (0.17,0.75) | 150 (80.2) | 0.37 (0.16,0.88) |
| Neither deficit nor surplus (balance) | 208 | 247 | 70 (33.7) | 0.24 (0.14,0.43) | 69 (27.9) | 0.24 (0.13,0.45) | 130 (62.5) | 0.13 (0.06,0.28) | 132 (53.4) | 0.12 (0.05,0.27) |
| Sufficient funds most of the time | 56 | 60 | 33 (58.9) | 0.80 (0.37,1.74) | 28 (46.7) | 0.64 (0.28,1.46) | 41 (73.2) | 0.19 (0.08,0.5) | 29 (48.3) | 0.1 (0.04,0.26) |
| **Marital status** | | | | | | | | | | |
| Married | 269 | 505 | 109 (40.5) | 1.0 (ref.) | 195 (38.6) | 1.0 (ref.) | 193 (71.7) | 1.0 (ref.) | 322 (63.8) | 1.0 (ref.) |
| Widowed | 301 | 66 | 163 (54.2) | 0.64 (0.41,1) | 48 (72.7) | 1.58 (0.81,3.05) | 230 (76.4) | 0.75 (0.47,1.18) | 58 (87.9) | 1.98 (0.85,4.64) |
| Unmarried or never married | 4 | 2 | 1 (25) | 0.40 (0.03,5.11) | 1 (50) | 3.01 (0.09,100.39) | 4(100) | 0 (0,) | 2 (100) | – |
| **Living status** | | | | | | | | | | |
| With family | 556 | 570 | 257 (46.2) | 1.0 (ref.) | 242 (42.5) | 1.0 (ref.) | 409 (73.6) | 1.0 (ref.) | 379 (66.5) | 1.0 (ref.) |
| With relatives or non-relatives | 6 | 1 | 5 (83.3) | 6.97 (0.6,81.26) | 1 (100) | – | 6(100) | 6.27 (1.65,23.85) | 1 (100) | – |
| Living alone | 12 | 2 | 11 (91.7) | 30.2 (2.47,370) | 1 (50) | 1.78 (0.1,31.99) | 12(100) | 9.54 (1.09,83.27) | 2 (100) | – |
| **Sources of income** | | | | | | | | | | |

(*Continued*)

**Table 3.** (Continued)

| | Female, N = 574 | Male, N = 573 | Physical domain, below average (<50%) QoL, N = 517 (45%) | | | | Psychological domain, below average (<50%) QoL, N = 809 (70.5%) | | | |
|---|---|---|---|---|---|---|---|---|---|---|
| | | | Female, 273 (47.6%) | | Male, 244 (42.6%) | | Female, 427 (74.4%) | | Male, 382 (66.7%) | |
| Age group | N | N | n(%) | OR (95% CI)* | n(%) | OR (95% CI)* | n(%) | OR (95% CI)* | n(%) | OR (95% CI)* |
| Own income | 26 | 214 | 11 (42.3) | 1.0 (ref.) | 35 (16.4) | 1.0(ref.) | 16 (61.5) | 1.0 (ref.) | 99 (46.3) | 1.0 (ref.) |
| Support from family or children | 533 | 340 | 254 (47.7) | 3.05 (0.7,13.35) | 200 (58.8) | 3.6 (2.01,6.44) | 398 (74.7) | 1.01 (0.35,2.9) | 267 (78.5) | 4.63 (2.56,8.38) |
| Government help or other sources | 15 | 19 | 8 (53.3) | 1.42 (0.21,9.66) | 9 (47.4) | 1.4 (0.45,4.39) | 13 (86.7) | 0.4 (0.08,1.94) | 16 (84.2) | 4.55 (1.06,19.46) |

*odds ratio (95% confidence interval (CI)) adjusted for age, education, occupation, marital status, living status and sources of income.

[56,57]. Therefore, it is imperative to make appropriate strategies and policies and take various support programs, such as life skills education programs, focusing on older women to meet the challenges to improve health and well-being for ageing populations now and in the coming years.

Our study has several strengths. The first is the face-to-face data collection from a large sample. The sample consists of 50% of women. QoL was measured using the WHOQoL-BREF questionnaire, which is widely used in developed and developing countries [34]. The questionnaire was validated for its use in rural Bangladesh, and this validated questionnaire was used for the current research [21]. The study has several limitations: Firstly, data were collected from one district, and thus, the results may be limited at the national level. Whilst it represents the situation in the Narail district, and the rural population is very homogenous in Bangladesh, the study's results need to be extrapolated with caution to other rural parts of Bangladesh. Secondly, the self-reported responses warrant personal bias in answering questions. Reporting errors and different perceptions about their QoL are very likely dependent on the participant's level of knowledge. Thirdly, the subjectivity in reporting the availability of funds in the last 12 months differs from the appropriate measure of SES. Fourthly, data were also collected in 2017 which brings possibility of some social changes during the last 6–7 years. Finally, the QoL was measured using the modified version of WHOQoL-BREF [21] with 19 items. Although the modified version of WHOQoL-BREF was validated using Rasch analysis and proposed to be more suitable in measuring QoL in rural Bangladesh, the magnitude of average or below QoL was not directly comparable with those used the WHOQoL-BREF 26-item questionnaire.

## Conclusions

The study concludes that more than 50% of people had below-average QoL in each domain, and the minimum was in the psychological domain. Females had consistently below QoL in all domains. Among males, as the level of education increases, QoL in the psychological domain improves; those who did not have their income and relied on either family or government support had a higher proportion of below-average QoL in all domains, and the retired males who had better SES had higher QoL in the environmental domain.

On a positive note, the study found a strong correlation between engagement in income-generating activities and improved QoL in the social and environmental domains for females. These results underscore the potential benefits of promoting such activities among older adults, providing valuable insights for policymakers and healthcare professionals striving to enhance the health and well-being of this demographic.

**Table 4. Association of socio-demographic factors with quality-of-life domains (physical, psychological, social and environmental) of 1147 elderly people in a rural district Narail in Bangladesh.**

| | Female, N = 574 | Male, N = 573 | Social domain, below average (<50%) QoL, N = 375 (32.7%) | | | | Environmental domain, below average (<50%) QoL, N = 603 (52.6%) | | | |
|---|---|---|---|---|---|---|---|---|---|---|
| | | | Female, 200 (34.8%) | | Male, 175 (30.1) | | Female, 322 (56.1) | | Male, 281 (49.0) | |
| | N | N | n(%) | OR (95% CI)* | n(%) | OR (95% CI)* | n(%) | OR (95% CI)* | n(%) | OR (95% CI) * |
| Age group | | | | | | | | | | |
| 60–69 | 399 | 330 | 142 (35.6) | 1.0 (ref.) | 102 (30.9) | 1.0 (ref.) | 217 (54.4) | 1.0 (ref.) | 154 (46.7) | 1.0 (ref.) |
| 70–79 years | 131 | 184 | 37 (28.2) | 0.56 (0.34,0.93) | 56 (30.4) | 0.78 (0.49,1.24) | 76 (58) | 1.15 (0.71,1.87) | 100 (54.3) | 1.35 (0.86,2.12) |
| 80+ | 44 | 59 | 21 (47.7) | 1.16 (0.56,2.39) | 17 (28.8) | 0.64 (0.32,1.28) | 29 (65.9) | 1.53 (0.71,3.28) | 27 (45.8) | 0.94 (0.48,1.85) |
| Education level | | | | | | | | | | |
| No Education | 298 | 202 | 100 (33.6) | 1.0 (ref.) | 80 (39.6) | 1.0 (ref.) | 158 (53) | 1.0 (ref.) | 120 (59.4) | 1.0 (ref.) |
| Below school secondary certificate | 247 | 278 | 89 (36) | 1.33 (0.88,2.01) | 75 (27) | 0.67 (0.44,1.03) | 146 (59.1) | 2.44 (1.59,3.73) | 129 (46.4) | 0.89 (0.57,1.37) |
| School secondary certificate and above | 29 | 93 | 11 (37.9) | 1.29 (0.53,3.12) | 20 (21.5) | 0.72 (0.36,1.46) | 18 (62.1) | 4.32 (1.76,10.65) | 32 (34.4) | 1.31 (0.68,2.56) |
| Occupation | | | | | | | | | | |
| Engaged in income activities | 9 | 262 | 4 (44.4) | 1.0 (ref.) | 63 (24) | 1.0 (ref.) | 5 (55.6) | 1.0 (ref.) | 130 (49.6) | 1.0 (ref.) |
| House duties | 341 | 0 | 112 (32.8) | 0.61 (0.16,2.32) | 0 | - | 186 (54.5) | 0.96 (0.25,3.64) | 0 | – |
| Retired | 22 | 79 | 13 (59.1) | 1.81 (0.38,8.64) | 27 (34.2) | 1.64 (0.95,2.83) | 16 (72.7) | 2.13 (0.42,10.73) | 25 (31.6) | 0.47 (0.28,0.80) |
| Unable to work | 202 | 232 | 71 (35.1) | 0.68(0.18,2.6) | 85 (36.6) | 1.83(1.24,2.7) | 115 (56.9) | 1.06 (0.28,4.05) | 126 (54.3) | 1.21 (0.85,1.72) |
| Socioeconomic status | | | | | | | | | | |
| Insufficient funds for the whole year | 116 | 79 | 50 (43.1) | 1.0 (ref.) | 39 (49.4) | 1.0 (ref.) | 89 (76.7) | 1.0 (ref.) | 60 (75.9) | 1.0 (ref.) |
| Insufficient funds for some of the time | 194 | 187 | 68 (35.1) | 0.84 (0.5,1.42) | 71 (38) | 0.61 (0.34,1.08) | 123 (63.4) | 0.52 (0.29,0.91) | 130 (69.5) | 0.78 (0.42,1.46) |
| Neither deficit nor surplus (balance) | 208 | 247 | 54 (26) | 0.52 (0.3,0.89) | 54 (21.9) | 0.32 (0.18,0.57) | 84 (40.4) | 0.16 (0.09,0.29) | 74 (30) | 0.14 (0.08,0.26) |
| Sufficient funds most of the time | 56 | 60 | 28 (50) | 1.44 (0.7,2.96) | 11 (18.3) | 0.22 (0.09,0.53) | 26 (46.4) | 0.17 (0.08,0.37) | 17 (28.3) | 0.14 (0.06,0.31) |
| Marital status | | | | | | | | | | |
| Married | 269 | 505 | 83 (30.9) | 1.0 (ref.) | 162 (32.1) | 1.0 (ref.) | 140 (52) | 1.0 (ref.) | 247 (48.9) | 1.0 (ref.) |
| Widowed | 301 | 66 | 117 (38.9) | 1.45 (0.96,2.2) | 13 (19.7) | 0.33 (0.16,0.67) | 179 (59.5) | 1.16 (0.77,1.75) | 34 (51.5) | 0.84 (0.45,1.59) |
| Unmarried or never married | 4 | 2 | 0 | 0 | 0 | 0 | 3 (75) | 4.08 (0.39,43.04) | 0 | – |
| Living status | | | | | | | | | | |
| With family | 556 | 570 | 188 (33.8) | 1.0 (ref.) | 173 (30.4) | 1.0 (ref.) | 305 (54.9) | 1.0 (ref) | 278 (48.8) | 1.0(ref.) |
| With relatives or non-relatives | 6 | 1 | 3 (50) | 1.04 (0.18,6.09) | 1 (100) | (0,) | 5 (83.3) | 2.62 (0.27,25.19) | 1 (100) | – |
| Living alone | 12 | 2 | 9 (75) | 5.94 (1.25,28.34) | 1 (50) | 2.15 (0.11,41.41) | 12 (100) | – | 2 (100) | – |
| Sources of income | | | | | | | | | | |

*(Continued)*

**Table 4.** (Continued)

| | Female, N = 574 | Male, N = 573 | Social domain, below average (<50%) QoL, N = 375 (32.7%) | | | | Environmental domain, below average (<50%) QoL, N = 603 (52.6%) | | | |
|---|---|---|---|---|---|---|---|---|---|---|
| | | | Female, 200 (34.8%) | | Male, 175 (30.1) | | Female, 322 (56.1) | | Male, 281 (49.0) | |
| | N | N | n(%) | OR (95% CI)* | n(%) | OR (95% CI)* | n(%) | OR (95% CI)* | n(%) | OR (95% CI)* |
| Own income | 26 | 214 | 13 (50) | 1.0 (ref.) | 47 (22) | 1.0 (ref.) | 16 (61.5) | 1.0(ref.) | 89 (41.6) | 1.0(ref.) |
| Support from family or children | 533 | 340 | 183 (34.3) | 1.01 (0.35,2.9) | 119 (35) | 1.81 (1.04,3.16) | 296 (55.5) | 2.76 (0.83,9.14) | 180 (52.9) | 2.53 (1.44,4.43) |
| Government help or other sources | 15 | 19 | 4 (26.7) | 0.4 (0.08,1.94) | 9 (47.4) | 2.45 (0.79,7.63) | 10 (66.7) | 2.53 (0.44,14.51) | 12 (63.2) | 3.9 (1.16,13.17) |

*odds ratio (95% confidence interval (CI)) adjusted for age, education, occupation, marital status, living status and sources of income.

## Supporting information

**S1 Data. Quality of life older adults PLOS one.**
(XLSX)

**S2 Data. The original twenty-six items in the WHOQoL-BREF questionnaire, including domain names and item numbers (in brackets).**
(DOCX)

## Acknowledgments

Acknowledgement goes to Dr Jessica Sharpand Tialara Harris for contributing to the initial background and Dr Nazim Uddin for data preparation. Data collectors and project managers, including Md Rafiqul Islam, Md Mafiz Biswas and Md Sajibul Islam, are acknowledged for their hard work contacting participants and collecting door-to-door data. Finally, the author thanked the study participants for their voluntary participation.

## Author Contributions

**Conceptualization:** Fakir M. Amirul Islam.

**Data curation:** Fakir M. Amirul Islam.

**Formal analysis:** Fakir M. Amirul Islam.

**Investigation:** Fakir M. Amirul Islam.

**Methodology:** Fakir M. Amirul Islam.

**Writing – original draft:** Fakir M. Amirul Islam.

**Writing – review & editing:** Fakir M. Amirul Islam.

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
