## [Decision Letter · Decision Letter 0]

12 Nov 2024

PONE-D-24-44126Gender difference in domain-specific quality of life measured by WHOQoL-BREF questionnaire and their associated factors among older adults in a rural district in BangladeshPLOS ONE

Dear Dr. Islam,

Thank you for submitting your manuscript to PLOS ONE. After careful consideration, we feel that it has merit but does not fully meet PLOS ONE’s publication criteria as it currently stands. Therefore, we invite you to submit a revised version of the manuscript that addresses the points raised during the review process.

We look forward to receiving your revised manuscript.

Kind regards,

Binh Thang Tran, MPH, PHD

Academic Editor

PLOS ONE

**Journal Requirements:**

**Additional Editor Comments:**

Although reviewers have provided positive comments on your paper, I encourage the authors to address their comments in detail. Also, I suggest considering the following points to further enhance the paper.

Major points:

Introduction:

1. Focus on the scope of your works on QoL and how its distinction sexes. What social mechanisms are associated between factors and QoL in each sex group among elderly people are you seeking in your study.

2. What advantages of WHOQoL-BREF scale measures QoL among this population compared to other scales?

Please reconstruct the introduction to reflect for your research question and aims of study.

Methods:

1. Data from surveys since 2017, please acknowledge the limitations of study.

2. QoL: The QoL measure provides more robust justification for using this 50% cutoff point for QoL and cite relevant sources of evidence to support this cutoff point.

3. Statistical analysis:

Back your study framework, you are trying to seek identifying the difference between 2 sexes (sample size estimation, page No.8), but your statistical analysis was not convinced (Table 3 to table 5) it is contradictory both using continuous variables and binary variables.

I invite authors to revisit data analysis to reframe for this again to have more comprehensive results, exploring interaction analysis between gender and predictors to gain deeper insights. The current approach to using subgroup analysis may not be sufficient to answer your research question.

Discussion

4. It is lengthy, please concentrate on your main findings on how its distinct between sexes on QoL and its predictors. Offer specific recommendations based on your findings and future direction.

Reviewers' comments:

Reviewer's Responses to Questions

**Comments to the Author**

1. Is the manuscript technically sound, and do the data support the conclusions?

Reviewer #1: Partly

Reviewer #2: Yes

Reviewer #3: Yes

2. Has the statistical analysis been performed appropriately and rigorously? 

Reviewer #1: Yes

Reviewer #2: Yes

Reviewer #3: Yes

3. Have the authors made all data underlying the findings in their manuscript fully available?

Reviewer #1: Yes

Reviewer #2: Yes

Reviewer #3: Yes

4. Is the manuscript presented in an intelligible fashion and written in standard English?

Reviewer #1: Yes

Reviewer #2: Yes

Reviewer #3: Yes

5. Review Comments to the Author

**Reviewer #1: **The manuscript entitled “Gender difference in domain-specific quality of life measured by WHOQoL-BREF questionnaire and their associated factors among older adults in a rural district in Bangladesh” highlighted a significant correlation between engagement in income- generating activities and improved QoL in the social and environmental domains for elderly females in rural Bangladesh. This finding offers valuable insights for healthcare professionals seeking to improve the overall health and well-being of the elderly population.

However, the manuscript needs further improvement. Please see details in the specific comments below.

Specific comments

Major concern

1. In title of this manuscript: Author didnot use the original of WHOQoL-BREF questionnaire, then should you revise this title with “modified WHOQoL-BREF scale”?

2. In the Methods section, the author should clarify whether the score of the modified WHOQoL-BREF (19 items) using Rasch analysis is equivalent to the original WHOQoL scale (26 items).

3. How to access SES variable in this manuscript is not appropriate for measuring of SES. Do authors have any reason to explain?

4. Authors showed in table 4 that among males, the prevalence of below average QoL in psychological domain odds ratio (OR) 1459774202.83, 95% confidence interval (CI)): 0, was higher among those who were unmarried or never married compared to those who married. Do you have any reason for this results? How is the goodness of fit in multivariable logistic regression which authors presented?

Minor comments

5. There are some minor writing issues in the manuscript.

6. In Tables 2 and 3, the authors should provide the full name of the categorized SES and source of income variables, ensuring consistency with the methodology outlined on page 9 (independent variables).

**Reviewer #2:** First impressions

Overall, the research design is sound and the writing is well-structured. The background section is excessively lengthy and requires condensation. This study has provided valuable insights for policy makers and healthcare professionals to implement appropriate intervention programs to improve the QoL of older adults. The paper was well structured with the appropriate language used. The study results addressed the research question and objective. Notwithstanding, there are some specific issues within each section as detailed in the comments below.

Introduction: Material hardship among older adults

The introduction was well-structured and provides comprehensive information regarding the paper, as well as was well organized and explained the importance of the study. Some points should be considered to improve the introduction:

The introduction is too long, the author should shorten it to suit the Objective of the research topic (About 2 papers A4).

Line 9 of Page 4: Authors should clarify before using abbreviations. For example: Quality of Life (QoL); And should be consistent in using QoL or QOL.

Line 18 of Page 4: Ref. 12 is too old. The author should consider citing appropriate references.

Methodology

- The author should review and unify the structure of the Research Methods section according to the guideline of Journal and consider merging the two contents Design and sampling & Sample size and statistical power.

- Statistical analysis:

Authors should provide additional information:

+ Descriptive statistics: Applied to describe data about research subjects' information according to fall risk status: Number, rate (%); Mean (Standard Error) for variables following normal distribution; Median (range) for variables that do not follow a normal distribution.

+ Inferential statistics: statistically significant with: *** p �0.001; ** p �0.01; * p � 0.05. Univariate and multivariate regression analysis to learn some factors related to the risk of falls in the elderly (statistically significant with 95%CI and *** p �0.001; ** p �0.01; * p � 0.05).

+ The potential confounding variables to correct the results of the relevant Tables.

Results and discussion

Footnote of Table 5: Authors should add adjusted variables.

Furthermore, authors should explain the abbreviations in the tables in each Footnote at the end of each table (Table 3-5).

Conclusion

Acceptable.

References, tables and figures

The format of the references is inconsistent with and does not follow the journal's guidelines.

References should be cited at the end of each sentence and Authors should follow the Journal's reference citation requirements.

Conclusion

Accept

**Reviewer #3:** A very interesting study on quality of life. The gender aspect is a very interesting aspect to explore in this field and the findings from the study are noteworthy. Especially on the marital status of women and the income status of men.

But I feel that the author's Introduction is quite rambling, and needs to go directly to the quality of life and gender issues. The author should consider shortening this part.

Besides, the study was conducted in only one district so I am quite concerned about the representativeness of this district for the whole rural area of Bangladesh. Of course the author also mentioned "in a rural district" in the title and this is not a big problem for such a good study.

The age group comparisons in the article are not appropriate as there are some comparisons with adult subjects while this study focused on the elderly. The author should replace the references appropriately.

The division of marriage and age groups seems inappropriate as the 60-69 group accounts for a large proportion. Age can be considered as a continuous variable in multivariate analysis.

Interviewing on quality of life requires consistency among the interview team, the author needs to briefly supplement the interview process, interviewer selection, training and control of possible errors.

I do not have many comments on this manuscript and wish the authors to be approved for publication soon.

6. PLOS authors have the option to publish the peer review history of their article (what does this mean?). If published, this will include your full peer review and any attached files.

Reviewer #1: No

Reviewer #2: No

Reviewer #3: No

---

## [Author Response · Author response to Decision Letter 0]

21 Nov 2024

The editor and the reviewers' comments are attached in one document.

---

## [Decision Letter · Decision Letter 1]

9 Dec 2024

PONE-D-24-44126R1Gender difference in domain-specific quality of life measured by modified WHOQoL-BREF questionnaire and their associated factors among older adults in a rural district in BangladeshPLOS ONE

Dear Dr. Islam,

Thank you for submitting your manuscript to PLOS ONE. After careful consideration, we feel that it has merit but does not fully meet PLOS ONE’s publication criteria as it currently stands. Therefore, we invite you to submit a revised version of the manuscript that addresses the points raised during the review process.

We look forward to receiving your revised manuscript.

Kind regards,

Binh Thang Tran, MPH, PHD

Academic Editor

PLOS ONE

Journal Requirements:

Additional Editor Comments:

Thank you for your excellent response to editor and reviewers. Most of the responses are reasonably accepted. However, I have further inquiries about the revised manuscript before it is formally published.

Introduction

Please narrow down the introduction, focusing on the research question. The current form is still quite lengthy.

Cite only the most relevant references in the introduction. The current version with 52 citations is excessive.

Method

Outcome defines:

• “….Further, the revised model did not show any sign of local dependency. Two items are about general health and overall QoL. Each item scores 1 to 5 on a Likert scale where 1 represent”:

The new paragraph about the revised model and item scoring is confusing. Please rephrase it for better understanding.

• modified WHOQoL-BREF is not well described in this section; what we need to see in this section is briefly about the modified version, and you just need to cite previous work (PMC5588978). Table 1 was not the result of this paper; please exclude it in this manuscript, or you can add it supplementary.

Discussion

Please be cautious when comparing results to other studies. Since your revised 26-item questionnaire was applied only to the Bangladeshi population, direct comparisons may not be accurate. Consider discussing the potential discrepancies and limitations of such comparisons.

Shorten your discussion section

Authorship: This is mostly your contribution; what others in the local or research team have not acknowledged and been authored in this work (PMC5588978, PMCPMC6451264)?

Reviewers' comments:

Reviewer's Responses to Questions

**Comments to the Author**

1. If the authors have adequately addressed your comments raised in a previous round of review and you feel that this manuscript is now acceptable for publication, you may indicate that here to bypass the “Comments to the Author” section, enter your conflict of interest statement in the “Confidential to Editor” section, and submit your "Accept" recommendation.

Reviewer #1: All comments have been addressed

Reviewer #2: All comments have been addressed

2. Is the manuscript technically sound, and do the data support the conclusions?

Reviewer #1: Yes

Reviewer #2: (No Response)

3. Has the statistical analysis been performed appropriately and rigorously? 

Reviewer #1: Yes

Reviewer #2: Yes

4. Have the authors made all data underlying the findings in their manuscript fully available?

Reviewer #1: Yes

Reviewer #2: Yes

5. Is the manuscript presented in an intelligible fashion and written in standard English?

Reviewer #1: Yes

Reviewer #2: Yes

6. Review Comments to the Author

Reviewer #1: The authors accordingly revised the manuscript regarding the reviewer's comments.

I am satisfied with the authors’ response and performed corrections.

Reviewer #2: The reviewer considers that the manuscript was improved.

Overall, this is a clear, concise, and well-written manuscript.

7. PLOS authors have the option to publish the peer review history of their article (what does this mean?). If published, this will include your full peer review and any attached files.

Reviewer #1: No

Reviewer #2: No

---

## [Author Response · Author response to Decision Letter 1]

10 Dec 2024

The response letter is attached.

---

## [Editor Report · Decision Letter 2]

22 Dec 2024

Gender difference in domain-specific quality of life measured by modified WHOQoL-BREF questionnaire and their associated factors among older adults in a rural district in Bangladesh

PONE-D-24-44126R2

Dear Dr. Islam,

We’re pleased to inform you that your manuscript has been judged scientifically suitable for publication and will be formally accepted for publication once it meets all outstanding technical requirements.

Kind regards,

Binh Thang Tran, MPH, PHD

Academic Editor

PLOS ONE

Additional Editor Comments (optional):

All my concerns have been well addressed. Congratulation !
---

## [Editor Report · Acceptance letter]

27 Dec 2024

PONE-D-24-44126R2 

PLOS ONE

Dear Dr. Islam, 

I'm pleased to inform you that your manuscript has been deemed suitable for publication in PLOS ONE. Congratulations! Your manuscript is now being handed over to our production team.

Kind regards, 

on behalf of

Dr. Binh Thang Tran 

Academic Editor

PLOS ONE